# The ΔPv-aCO$_2$/ΔCa-vO$_2$ ratio as a predictor of mortality in patients with severe acute respiratory distress syndrome related to COVID-19

**Jesús Salvador Sánchez Díaz**[1], **Karla Gabriela Peniche Moguel**[1]*, **José Manuel Reyes-Ruiz** [2]*, **Orlando Rubén Pérez Nieto**[3], **Diego Escarramán Martínez**[4], **Eder Iván Zamarrón López**[5], **María Verónica Calyeca Sánchez**[1]

**1** Critical Care Department, Unidad Médica de Alta Especialidad, Hospital de Especialidades No. 14, Centro Médico Nacional "Adolfo Ruiz Cortines", Instituto Mexicano del Seguro Social (IMSS), Veracruz, Mexico, **2** Department of Health Research, Unidad Médica de Alta Especialidad, Hospital de Especialidades No. 14, Centro Médico Nacional "Adolfo Ruiz Cortines", Instituto Mexicano del Seguro Social (IMSS), Veracruz, Mexico, **3** Critical Care Department, Hospital General San Juan del Río, Secretaría de Salud, Querétaro, Mexico, **4** Department of Anesthesiology, Centro Médico Nacional "La Raza", Instituto Mexicano del Seguro Social (IMSS), Mexico City, Mexico, **5** Critical Care Department, Hospital General Regional No. 6, Instituto Mexicano del Seguro Social (IMSS), Ciudad Madero, Tamaulipas, Mexico

* jose.reyesr@imss.gob.mx (JMRR); gabrielapenichemd@gmail.com (KGPM)

**Data Availability Statement:** Data are available from the local Ethics and Research Committee of

## Abstract

### Objective

To evaluate the central venous-to-arterial carbon dioxide difference combined with arterial-to-venous oxygen content difference (ΔPv-aCO2/ΔCa-vO2 ratio) as a predictor of mortality in patients with COVID-19-related severe acute respiratory distress syndrome (ARDS).

### Methods

Patients admitted to the intensive care unit with severe ARDS secondary to SARS-CoV-2, and invasive mechanical ventilation were included in this single-center and retrospective cohort study performed between April 18, 2020, and January 18, 2022. The tissue perfusion indexes (lactate, central venous oxygen saturation [ScvO2], and venous-to-arterial carbon dioxide pressure difference [ΔPv-aCO2]), anaerobic metabolism index (ΔPv-aCO2/ΔCa-vO2 ratio), and severity index (Simplified Acute Physiology Score II [SAPSII]) were evaluated to determine its association with the mortality through Cox regression analysis, Kaplan-Meier curve and receiver operating characteristic (ROC) curve.

### Results

One hundred fifteen patients were included in the study and classified into two groups, the survivor group (n = 54) and the non-survivor group (n = 61). The lactate, ScvO$_2$, ΔPv-aCO$_2$, and ΔPv-aCO$_2$/ΔCa-vO$_2$ ratio medians were 1.6 mEq/L, 75%, 5 mmHg, and 1.56 mmHg/mL, respectively. The ΔPv-aCO$_2$/ΔCa-vO$_2$ ratio (Hazard Ratio (HR) = 1.17, 95% confidence interval (CI) = 1.06–1.29, $p$ = 0.001) was identified as a mortality biomarker for patients with

the Unidad Médica de Alta Especialidad No. 14, IMSS (contact via luis.jimenezlo@imss.gob.mx) for researchers who meet the criteria for access to confidential data.

**Funding:** The author(s) received no specific funding for this work.

**Competing interests:** The authors have declared that no competing interests exist.

COVID-19-related severe ARDS. The area under the curve for ΔPv-aCO$_2$/ΔCa-vO$_2$ ratio was 0.691 (95% CI 0.598–0.774, $p = 0.0001$). The best cut-off point for ΔPv-aCO$_2$/ΔCa-vO$_2$ ratio was >2.14 mmHg/mL, with a sensitivity of 49.18%, specificity of 85.19%, a positive likelihood of 3.32, and a negative likelihood of 0.6. The Kaplan-Meier curve showed that survival rates were significantly worse in patients with values greater than this cut-off point.

## Conclusions

The ΔPv-aCO$_2$/ΔCa-vO$_2$ ratio could be used as a predictor of mortality in patients with severe ARDS secondary to SARS-CoV-2.

## Introduction

Most monitoring of critically ill patients maintains an interest in macrohemodynamic variables [1]. On the other hand, the gasometric analysis provides a formal assessment of tissue perfusion [2] and anaerobic metabolism [3] through serum lactate, central venous oxygen saturation (ScvO$_2$), venous-to-arterial carbon dioxide pressure difference (ΔPv-aCO$_2$), and central venous-to-arterial carbon dioxide difference combined with arterial-to-venous oxygen content difference (ΔPv-aCO$_2$/ΔCa-vO$_2$ ratio). The above is interesting because it has been documented that patients with Coronavirus disease 2019 (COVID-19) have alterations in tissue perfusion [4] and oxygen metabolism [5]. Serum lactate is the most frequently used marker of tissue perfusion [6], increasing in the presence of cellular hypoxia or low peripheral perfusion [7]; a level >2 mmol/L is the most commonly used cut-off point [8]. ScvO$_2$ surrogates the ratio of oxygen consumption/oxygen availability (VO$_2$/DO$_2$), reliably reflecting global cellular oxygenation [9]. The reference value of ScvO2 is 70%; in pathological situations, this value may increase or decrease [10]. ScvO$_2$ should be analyzed based on its determinants: arterial oxygen saturation (SaO$_2$), oxygen transport (hemoglobin), oxygen availability (DO$_2$), and oxygen consumption (VO$_2$) [11].

The ΔPv-aCO$_2$ is a good indicator of venous blood flow in peripheral tissues [12]. When blood flow is appropriate (ideal cardiac output), CO$_2$ will be well removed, and the ΔPv-aCO$_2$ will be ≤ 6 mmHg; but without proper blood flow, CO$_2$ will be poorly removed, and the ΔPv-aCO$_2$ will be >6 mmHg (non-ideal cardiac output) [13]. Some factors can modify ΔPv-aCO$_2$, such as hyper- or hypoventilation, hypo- or hyperoxemia, fever or hypothermia, decreased or increased hemoglobin, and deficit or excess hydrogen ions, which should be considered [14].

The ΔPv-aCO$_2$/ΔCa-vO$_2$ ratio can surrogate the respiratory quotient (RQ), representing the VCO$_2$/VO$_2$ ratio (carbon dioxide production/oxygen consumption). In anaerobic conditions, VCO$_2$ exceeds VO$_2$ resulting in an RQ >1. The RQ will increase by higher VCO$_2$ or lower VO$_2$, reflecting hypoxic or cytopathic hypoxia; consequently, the anaerobic metabolism highlights the usefulness of the ΔPv-aCO$_2$/ΔCa-vO$_2$ ratio as a surrogate for RQ [15]. We must also consider variables that modify ΔPv-aCO$_2$/ΔCa-vO$_2$ ratio (CO$_2$, oxygen, temperature, hemoglobin, and hydrogen ions), which are the same as ΔPv-aCO$_2$.

In patients with Severe Acute Respiratory Distress Syndrome (ARDS) secondary to Severe Acute Respiratory Syndrome Coronavirus 2 (SARS-CoV-2), tissue perfusion is altered, and severe hypoxemia [16] compromises the VO$_2$/DO$_2$ ratio, increasing anaerobic metabolism, which, if not corrected will cause dysoxia and finally cell death [17]. Hence, ΔPv-aCO$_2$/ΔCa-vO$_2$ ratio could help predict mortality in patients with severe ARDS secondary to SARS-CoV-2.

## Material and methods

### Study design and patients

A single-center, retrospective cohort study was conducted in the Intensive Care Unit (ICU) of the Unidad Médica de Alta Especialidad, Hospital de Especialidades No. 14, Centro Médico Nacional "Adolfo Ruiz Cortines" of the Instituto Mexicano del Seguro Social (IMSS), Veracruz, Mexico from April 18, 2020, to January 18, 2021. Convenience sampling was performed, which included patients admitted to the ICU with ARDS secondary to SARS-CoV-2. Inclusion criteria were: (1) age >18 years, (2) any gender, (3) confirmed SARS-CoV-2 infection by reverse transcriptase polymerase chain reaction (RT-PCR), and (4) severe ARDS (PaO$_2$/FiO$_2$ ≤100 mmHg) defined according to Berlin criteria [18] with invasive mechanical ventilation (IVM). (1) Patients with diseases that could affect the hemoglobin or CO$_2$ levels such as hematologic diseases, chronic obstructive pulmonary disease (COPD), known neuromuscular disease or known hyperbaric respiratory failure; (2) patients with an incomplete variable registry; or (3) pregnant were excluded from this study. All patients were intubated in the ICU and some of them received norepinephrine (n = 26, 22.60%) as the only vasopressor. These patients did not require ionotropic support other than the vasoconstrictor norepinephrine. All the patients were sedated using propofol and mechanical ventilation was started. Lung-protective mechanical ventilation was applied in the volume assist-controlled mode using the Puritan Bennett 840 ventilator (Medtronic; Carlsbad, California, USA), with the following settings: tidal volume of 6 mL/Kg predicted body wight, plateau pressure ≤27 cmH$_2$O, and driving pressure ≤15 cmH$_2$O. After 30 min of ventilation in a supine positioning the ventilatory variables, including perfusion indexes and anaerobic metabolism, were assessed. The arterial and central venous blood gases were determined in the GEM® PREMIER™4000 with iQM® equipment.

The propofol infusion was administered to maintain a Richmond Agitation-Sedation Scale (RASS) score of -3 (moderate sedation; the patient had any movement in response to voice, but no eye contact) and overcome ventilator asynchrony, obtain a level of awake sedation optimizing the patient´s respiratory status without effects on respiratory pattern, respiratory drive, and arterial and central venous blood gases.

### Data collection

All data from the patients meeting the inclusion criteria were collected from the electronic medical records. A single physician specializing in critical care collected all the data, taking them from the clinical record. The variables obtained were classified into demographic (gender, age, body mass index [BMI]), comorbidities (diabetes mellitus, systemic arterial hypertension [SAH], smoking, chronic kidney disease [CKD], cardiopathy), gasometrical (hydrogen potential [pH], arterial oxygen pressure/inspired oxygen fraction [PaO$_2$/FiO$_2$], arterial carbon dioxide pressure [PaCO$_2$], bicarbonate [HCO$_3$-], base), tissue perfusion indexes (lactate, central venous oxygen saturation [ScvO$_2$], and venous-to-arterial carbon dioxide pressure difference [ΔPv-aCO$_2$]), anaerobic metabolism index (central venous-to-arterial carbon dioxide difference combined with arterial-to-venous oxygen content difference [ΔPv-aCO$_2$/ΔCa-vO$_2$ ratio]), and severity index (Simplified Acute Physiology Score II [SAPSII]). Other variables such as creatinine, D-dimer, C-reactive protein, fibrinogen, glutamic oxaloacetic transaminase [GOT], glutamic pyruvic transaminase [GPT], and vasopressor were also included in this study. According to the clinical records, variables were obtained once the ICU admitted patients in a supine position after intubation (within the first 30 minutes).

## Definitions

The perfusion indexes and anaerobic metabolism were calculated according to the following formulas:

$$CaO_2 = (1.34 \times SaO_2 \times hemoglobin) + (0.003 \times PaO_2)$$

$$CvO_2 = (1.34 \times SvO_2 \times hemoglobin) + (0.003 \times PvO_2)$$

$$\Delta Pv{-}aCO_2 = PcvCO_2 - PaCO_2$$

$$\Delta Pv{-}aCO_2/\Delta Ca - vO_2 \text{ ratio} = \Delta Pv{-}aCO_2/\Delta Ca - vO_2$$

## Statistical analysis

Data are expressed as number (%) for categorical and as mean (standard deviation, ±SD) or median (interquartile range, IQR) for continuous variables. Data distribution was analyzed with the Kolmogorov-Smirnov test, histograms, and Q-Q plots. The Mann-Whitney U test was used to compare numerical variables with no normal or non-parametric distribution. A Student's t-test compared numerical variables with a normal or parametric distribution. The association between categorical variables was determined with the chi-square test ($x^2$) or Fisher's exact test according to cross-table assumptions. A Cox regression analysis measured the mortality as the dependent variable, adjusted with perfusion (serum lactate, ScvO$_2$, and $\Delta$Pv-aCO$_2$) and anaerobic metabolism ($\Delta$Pv-aCO$_2$/$\Delta$Ca-vO$_2$ ratio) variables. Results are summarized as a Hazard Ratio (HR) and 95% confidence intervals (95% CI). A Hosmer-Lemeshow adjustment (p> 0.05) assessed the calibration. Receiver Operator Characteristic (ROC) curves were performed to evaluate and compare the Area Under the Curve (AUC) of $\Delta$Pv-aCo$_2$/$\Delta$Ca-vO$_2$ and SAPS II associated with COVID-19 mortality. The optimal cut-off points were determined considering the Youden index by showing the trade-off- between sensitivity and specificity. A Kaplan-Meier survival analysis compared both groups to the established $\Delta$Pv-aCo$_2$/$\Delta$Ca-vO$_2$ and SAPS II optimal cut-off points. The correlation between $\Delta$Pv-aCO$_2$/$\Delta$Ca-vO$_2$ and SAPS II was calculated using the Spearman correlation test. A p-value < 0.05 was considered a statistically significant difference. Data analysis was performed using R Studio v4.03 Statistical (R Foundation, Vienna, Austria), MedCalc Statistical Software (Ostend, Belgium), and SPSS v.25 Software (IBM, New York, USA).

## Ethics

The present study was conducted according to the Strengthening the Reporting of Observational Studies in Epidemiology (STROBE) methodology for observational studies [19]. The study protocol was approved (register number R-2021-3001-061) by the local Ethics and Research Committee of the Unidad Médica de Alta Especialidad No. 14, IMSS, including the exemption of the requirement for informed consent. All patients included were provided with identity protection through the assignment of an identification number and were also closely followed up until there was an outcome. Moreover, this study was compliant with the Declaration of Helsinki. We certify that all protocols and methods follow relevant guidelines and regulations.

## Results

### Patient characteristics

This study cohort included one hundred and fifteen subjects (Fig 1). Patients were stratified into survivor (n = 54) and non-survivor (n = 61). The median age was 65 years (57.5–73). The

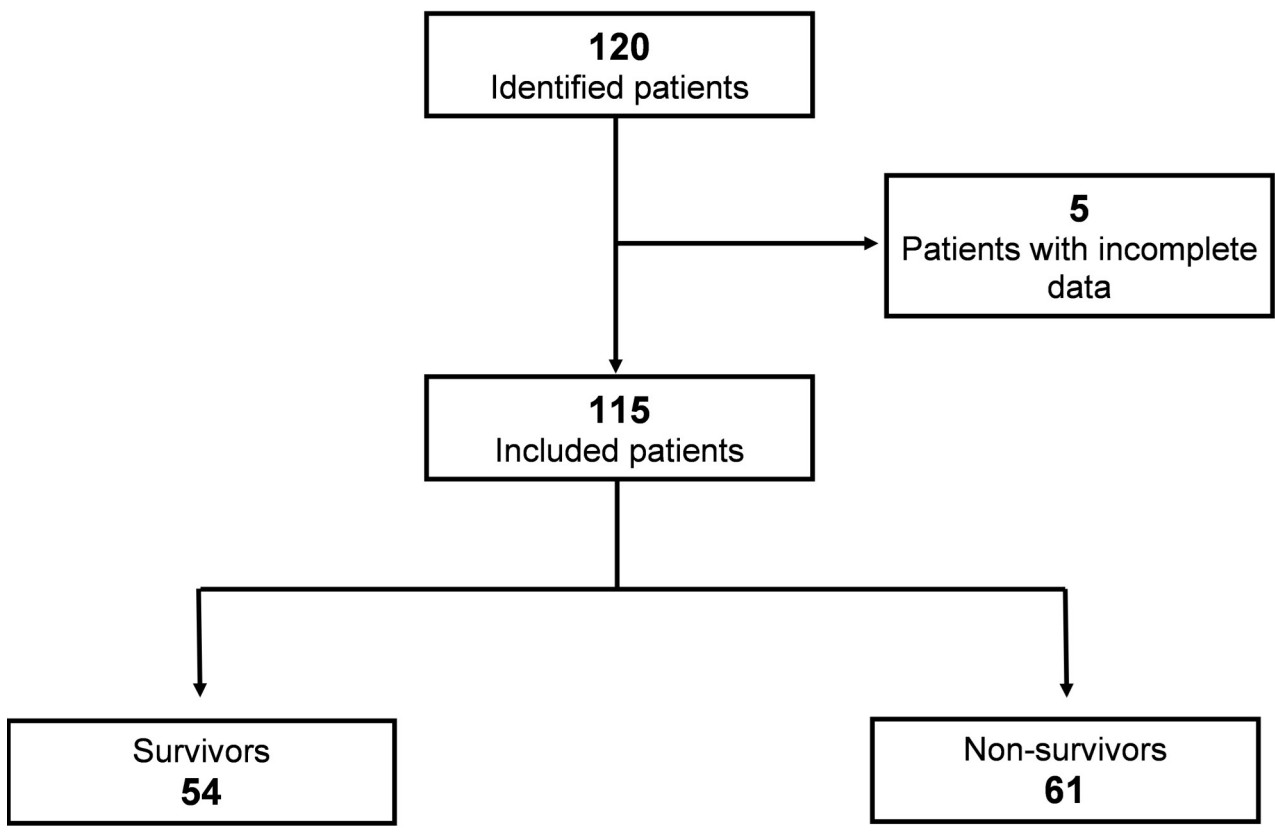

**Fig 1. Flowchart of hospitalized patients included in the cohort and their outcome.** Signs, symptoms, and radiological findings suggested COVID-19; however, SARS-CoV-2 infection in all patients was confirmed by a positive RT-PCR from a nasal/throat swab.

predominant gender was male (61.73%), the most frequent comorbidity was SAH (67.82%), and the mean SAPS II was 75.35 points (SD ± 9.26). The medians for lactate, ScvO$_2$, and $\Delta$Pv-aCO$_2$ were 1.6 mEq/L (1.2–2.1), 75% (68.5–81), and 5 mmHg (3–9), respectively. The median anaerobic metabolism index or $\Delta$Pv-aCO$_2$/$\Delta$Ca-vO$_2$ ratio was 1.56 mmHg/mL (1.02–2.67). The remainder is summarized in Table 1.

## Risk factors for mortality in patients with severe ARDS caused by COVID-19

Table 2 show the Cox regression analysis for mortality in patients with severe ARDS secondary to SARS-CoV-2. In the univariate Cox regression analysis, the variables BMI, smoking, diabetes, SAPS II, vasopressor, pH, base, $\Delta$Pv-aCO$_2$, and $\Delta$Pv-aCO$_2$/$\Delta$Ca-vO$_2$ ratio had statistical significance. In the multivariate Cox regression analysis, BMI (HR 1.04, 95% CI 1.01–1.08, $p$ = 0.007), SAPS II (HR 1.04, 95% CI 1.01–1.08, $p$ = 0.005), and $\Delta$Pv-aCO$_2$/$\Delta$Ca-vO$_2$ ratio (HR 1.17, 95% CI 1.06–1.29, $p$ = 0.001) maintained statistical significance. Fig 2 shows the forest plot of final Cox regression model for mortality in patients with severe ARDS related to COVID-19.

## The $\Delta$Pv-aCO$_2$/$\Delta$Ca-vO$_2$ ratio as an independent predictor of survival in patients with severe ARDS related to COVID-19

The AUC for $\Delta$Pv-aCO$_2$/$\Delta$Ca-vO$_2$ ratio was 0.691(95% CI 0.598–0.774, $p$ = 0.0001), with the best cut-off point of >2.14 mmHg/mL (sensitivity 49.18%, specificity 85.19%, positive

**Table 1. Demographics and clinical characteristics of COVID-19 patients according to survival.**

| Variable | Overall (n = 115) | Survivors (n = 54) | Non-survivors (n = 61) | P-value |
|---|---|---|---|---|
| **Demographic** | | | | |
| Age, years | 65 (57.5–73) | 64 (50.25–73.5) | 67 (60–72) | 0.127 |
| Male, n (%) | 71 (61.73%) | 33 (61.11%) | 38 (62.29%) | 1 |
| BMI, Kg/m² | 33.30 (29.39–36.31) | 32.77 (27.89–34.83) | 33.95 (30.12–37.78) | 0.1 |
| **Comorbidity** | | | | |
| Smoking, n (%) | 39 (33.91%) | 11 (20.37%) | 28 (45.90%) | **0.007** |
| Diabetes, n (%) | 61 (53.04%) | 23 (42.59%) | 38 (62.29%) | 0.054 |
| Hypertension, n (%) | 78 (67.82%) | 32 (59.25%) | 46 (75.40%) | 0.098 |
| CKD, n (%) | 8 (6.95%) | 2 (3.7%) | 6 (9.83%) | 0.279 |
| Cardiopathy, n (%) | 9 (7.82%) | 4 (7.4%) | 5 (8.19%) | 1 |
| **Clinical and laboratory data** | | | | |
| SAPS II, points | 75.35 (± 9.26) | 72 (± 8.15) | 78.31 (± 9.24) | **<0.0005** |
| Vasopressor, n (%) | 26 (22.60%) | 4 (7.40%) | 22 (36.06%) | **<0.0005** |
| Temperature, ˚C | 36.70 (36.40–37) | 36.65 (36.40–37.08) | 36.70 (36.40–36.90) | 0.787 |
| pH | 7.37 (7.28–7.43) | 7.38 (7.31–7.43) | 7.34 (7.23–7.43) | 0.150 |
| PaO₂/FiO₂, mmHg | 76 (61.5–94) | 85.5 (69.25–110.25) | 70 (59–88) | **0.004** |
| PaCO₂, mmHg | 41 (37–51) | 41 (38–50.75) | 43 (36–51) | 0.924 |
| HCO₃-, mEq/L | 24.37 (± 4.86) | 25.29 (± 4.88) | 23.55 (± 4.74) | 0.055 |
| Base, mEq/L | -1.74 (± 5.09) | -0.55 (± 4.77) | -2.79 (± 5.18) | **0.017** |
| Creatinine, mg/dL | 0.82 (0.62–1.1) | 0.7 (0.6–0.9) | 0.94 (0.73–1.4) | **<0.0001** |
| GOT, U/L | 34 (22.5–47.5) | 31 (19–44) | 36.5 (26.25–55.75) | 0.072 |
| GPT, U/L | 40 (28–55.5) | 41 (28–56) | 37 (28.5–51.25) | 0.511 |
| C-reactive protein, mg/dL | 107 (60–194) | 92 (54–139) | 153 (72–243) | **0.013** |
| D-dimer, ng/mL | 1607 (661–3354) | 1149 (584.8–3909.8 | 1702 (797–2491) | 0.342 |
| Fibrinogen, mg/L | 289 (260–303) | 279 (223–302.5) | 290 (270–308) | 0.052 |
| Hemoglobin, g/dL | 13.7 (12.3–14.9) | 13.7 (12.3–14.6) | 13.7 (12.6–14.9) | 0.58 |
| HbA1C, % | 6.4 (6–7.85) | 6.3 (5.9–7) | 6.9 (6.1–8.2) | **0.043** |
| Lactate, mEq/L | 1.6 (1.2–2.1) | 1.6 (1.2–2.17) | 1.6 (1.2–2) | 0.850 |
| ScvO₂, % | 75 (68.5–81) | 75 (69.5–79) | 76 (68–82) | 0.366 |
| ΔPv-aCO₂, mmHg | 5 (3–9) | 5 (3–6.75) | 6 (5–9) | **0.039** |
| ΔPv-aCO₂/ΔCa-vO₂ | 1.56 (1.02–2.67) | 1.35 (0.83–1.97) | 2.05 (1.38–3.60) | **<0.0005** |
| Days MV | 6 (4–9) | 5 (4–6) | 7 (4–10) | **0.006** |

Data are shown as number (%) for categorical and as mean ±SD or median (IQR) for continuous variables. Statistically significant *p* values (<0.05) are highlighted in bold. BMI, body mass index; CKD, chronic kidney disease; SAPS II, Simplified Acute Physiology Score II; pH, potential hydrogen; PaO₂/FiO₂, arterial oxygen pressure/inhaled oxygen fraction; PaCO₂, arterial carbon dioxide pressure; HCO₃-, bicarbonate; GOT, glutamic oxalacetic transaminase; GPT, glutamic pyruvic transaminase; HbA1C, glycosylated hemoglobin; ScvO₂, central venous oxygen saturation; ΔPv-aCO₂, venous-to-arterial carbon dioxide pressure difference; ΔPv-aCO₂/ΔCa-vO₂ ratio, central venous-to-arterial carbon dioxide difference combined with arterial-to-venous oxygen content difference; Days MV, days mechanical ventilation.

likelihood ratio (LR+) 3.32, and negative likelihood ratio (LR-) 0.6 (Fig 3A). The best cut-off point obtained by Youden's index for SAPS II was >74 points (AUC = 0.696 (95% CI 0.603–0.778, *p* = 0.0001), sensitivity 65.57%, specificity 64.81%, LR+ 1.86, and LR- 0.53) (Fig 3B). Fig 4A shows the Kaplan-Meier curve of the ΔPv-aCO₂/ΔCa-vO₂ ratio for 30-day survival, showing a statistically significant difference between survivors and non-survivors when the cut-off point of >2.14 mmHg/mL was used. Fig 4B shows the Kaplan-Meier curve of SAPS II for 30-day survival. The linear correlation between ΔPv-aCO₂/ΔCa-vO₂ ratio and SAPS II was $R = 0.21$ with $p = 0.025$ (Fig 5).

**Table 2. Univariate and multivariate cox regression analysis of mortality in patients with SARS-CoV-2-induced acute respiratory distress syndrome (ARDS).**

| Variable | Univariate | | | Multivariate | | |
|---|---|---|---|---|---|---|
| | HR | (95% CI) | P Value | HR | (95% CI) | P Value |
| Age | 1.01 | 0.99–1.04 | 0.079 | - | - | - |
| Male | 0.97 | 0.57–1.63 | 0.913 | - | - | - |
| BMI | 1.04 | 1.01–1.08 | **0.004** | 1.04 | 1.01–1.08 | **0.007** |
| Smoking | 2.06 | 1.24–3.42 | **0.005** | 1.63 | 0.92–2.86 | 0.089 |
| Diabetes | 1.69 | 1.01–2.83 | **0.044** | 1.57 | 0.90–2.76 | 0.110 |
| Hypertension | 1.76 | 0.97–3.21 | 0.061 | - | - | - |
| ERC | 1.57 | 0.67–3.66 | 0.291 | - | - | - |
| Cardiopathy | 0.52 | 0.12–2.12 | 0.363 | - | - | - |
| SAPS II | 1.05 | 1.02–1.08 | **0.0002** | 1.04 | 1.01–1.08 | **0.005** |
| Vasopressor | 2.76 | 1.62–4.68 | **<0.0005** | 1.71 | 0.89–3.28 | 0.101 |
| Temperature | 1.31 | 0.84–2.04 | 0.218 | - | - | - |
| pH | 0.07 | 0.007–0.777 | **0.030** | 0.22 | 0.01–4.90 | 0.342 |
| Pa$O_2$/Fi$O_2$ | 0.99 | 0.99–1 | 0.074 | - | - | - |
| Pa$CO_2$ | 0.99 | 0.98–1 | 0.714 | - | - | - |
| $HCO_3$- | 0.95 | 0.9–1 | 0.081 | - | - | - |
| Base | 0.94 | 0.89–0.98 | **0.014** | 1 | 0.93–1.07 | 0.933 |
| Creatinine | 1.07 | 0.99–1.16 | 0.067 | - | - | - |
| GOT | 1 | 0.99–1 | 0.444 | - | - | - |
| GPT | 1 | 0.99–1 | 0.340 | - | - | - |
| C-reactive protein | 1 | 0.99–1 | 0.668 | - | - | - |
| D-dimer | 1 | 0.99–1 | 0.055 | - | - | - |
| Fibrinogen | 1 | 0.99–1 | 0.197 | - | - | - |
| HbA1C | 1.03 | 0.89–1.20 | 0.626 | - | - | - |
| Lactate | 0.94 | 0.68–1.29 | 0.709 | - | - | - |
| Scv$O_2$ | 1.01 | 0.97–1.04 | 0.527 | - | - | - |
| $\Delta$Pv-a$CO_2$ | 1.05 | 1–1.10 | **0.040** | 0.97 | 0.90–1.04 | 0.484 |
| $\Delta$Pv-a$CO_2$/$\Delta$Ca-v$O_2$ | 1.11 | 1.05–1.18 | **<0.0005** | 1.17 | 1.06–1.29 | **0.001** |
| Days MV | 1.02 | 0.97–1.07 | 0.273 | - | - | - |

Candidate predictors with statistically significant differences ($p< 0.05$) in univariate Cox regression analysis were included in a multivariate Cox regression analysis. Hazard ratio (HR) and 95% Confidence Interval (95% CI) are reported. Statistically significant *P* values (<0.05) are highlighted in bold. BMI, body mass index; CKD, chronic kidney disease; SAPS II, Simplified Acute Physiology Score II; pH, potential hydrogen; Pa$O_2$/Fi$O_2$, arterial oxygen pressure/inhaled oxygen fraction; Pa$CO_2$, arterial carbon dioxide pressure; $HCO_3$-, bicarbonate; GOT, glutamic oxalacetic transaminase; GPT, glutamic pyruvic transaminase; HbA1C, glycosylated hemoglobin; Scv$O_2$, central venous oxygen saturation; $\Delta$Pv-a$CO_2$, venous-to-arterial carbon dioxide pressure difference; $\Delta$Pv-a$CO_2$/$\Delta$Ca-v$O_2$ ratio, central venous-to-arterial carbon dioxide difference combined with arterial-to-venous oxygen content difference; Days MV, days mechanical ventilation.

## Discussion

The priority in patients with ARDS secondary to SARS-CoV-2 will be to avoid dysoxia [20]. Circulatory homeostasis between macrocirculation, microcirculation, and the cell will maintain the flow of oxygen to the different organs avoiding tissue hypoxia, a condition that can cause cell damage and death [21]. Anaerobic metabolism occurs when $DO_2$ decreases to critical levels (< 7 ml/kg/min) concerning $VO_2$ by exhaustion of compensatory mechanisms [22]. Indirect markers such as Scv$O_2$, $\Delta$Pv-a$CO_2$, lactate, and $\Delta$Pv-a$CO_2$/$\Delta$Ca-v$O_2$ ratio can help assess $VO_2$/$DO_2$ ratio, tissue perfusion, and anaerobic metabolism [22–24]. We must consider that any parameter has limitations, but it is up to the physician to choose the best marker, contextualizing each patient, which makes multimodal monitoring imperative.

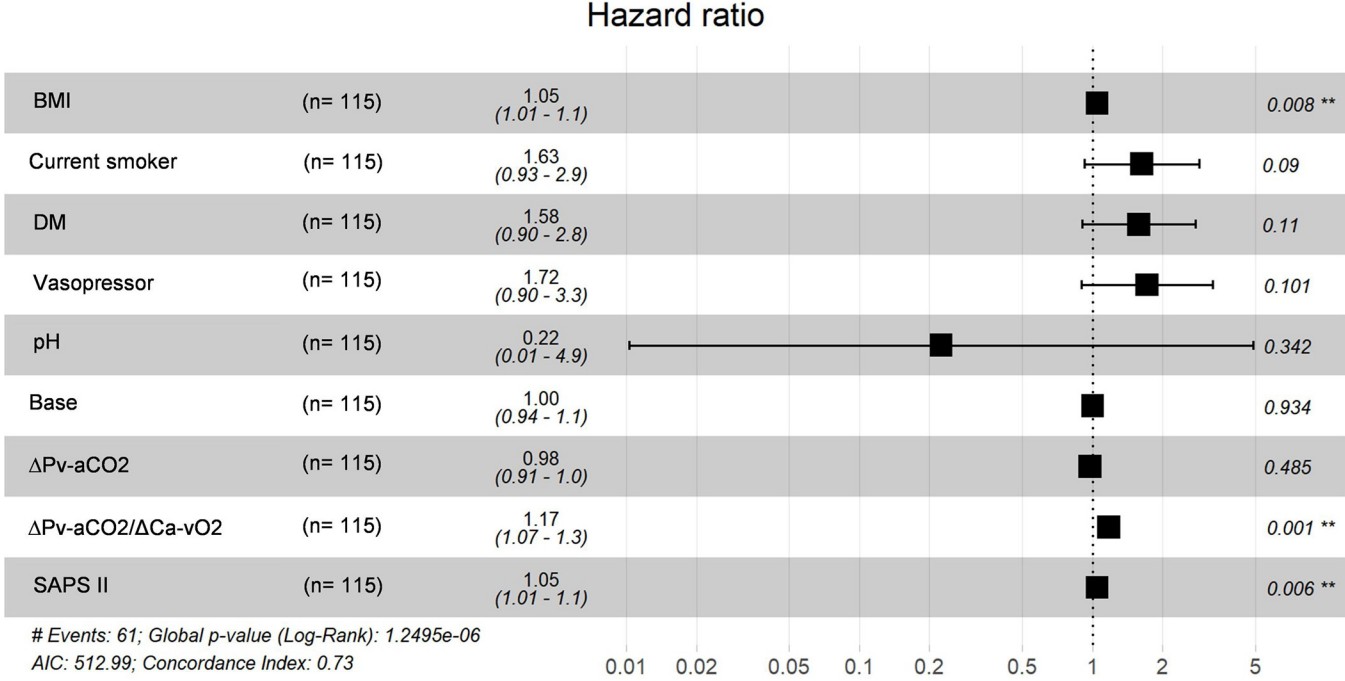

**Fig 2. Forest plot of multivariate COX regression analysis.** The squares represent the hazard ratio (HR), and the horizontal lines show the confidence Interval. Two asterisks indicate a significant difference at $p < 0.01$. BMI, body mass index; DM, diabetes mellitus; pH, potential hydrogen; $\Delta Pv\text{-}aCO_2$, arteriovenous oxygen pressure delta; $\Delta Pv\text{-}aCO_2/\Delta Ca\text{-}vO_2$ ratio, central venous-to-arterial carbon dioxide difference combined with arterial-to-venous oxygen content difference; SAPS II, Simplified Acute Physiology Score II.

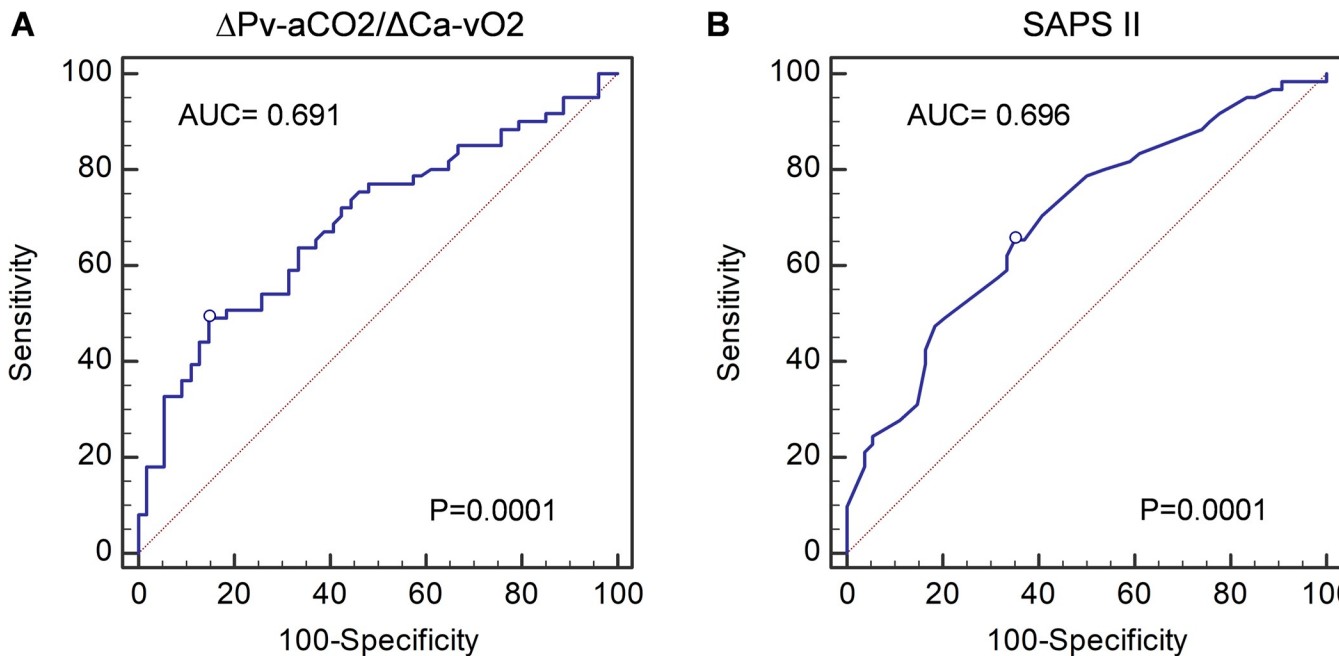

**Fig 3. The AUC of the $\Delta Pv\text{-}aCo_2/\Delta Ca\text{-}vO_2$ ratio and SAPS II.** Receiver Operating Characteristic (ROC) curves on sensitivity and specificity of (A) $\Delta Pv\text{-}aCo_2/\Delta Ca\text{-}vO_2$ and (B) SAPS II for predicting mortality in patients with severe ARDS due to SARS-CoV-2 infection. AUC, the area under the curve. A $p$-value of less than 0.05 was considered statistically significant.

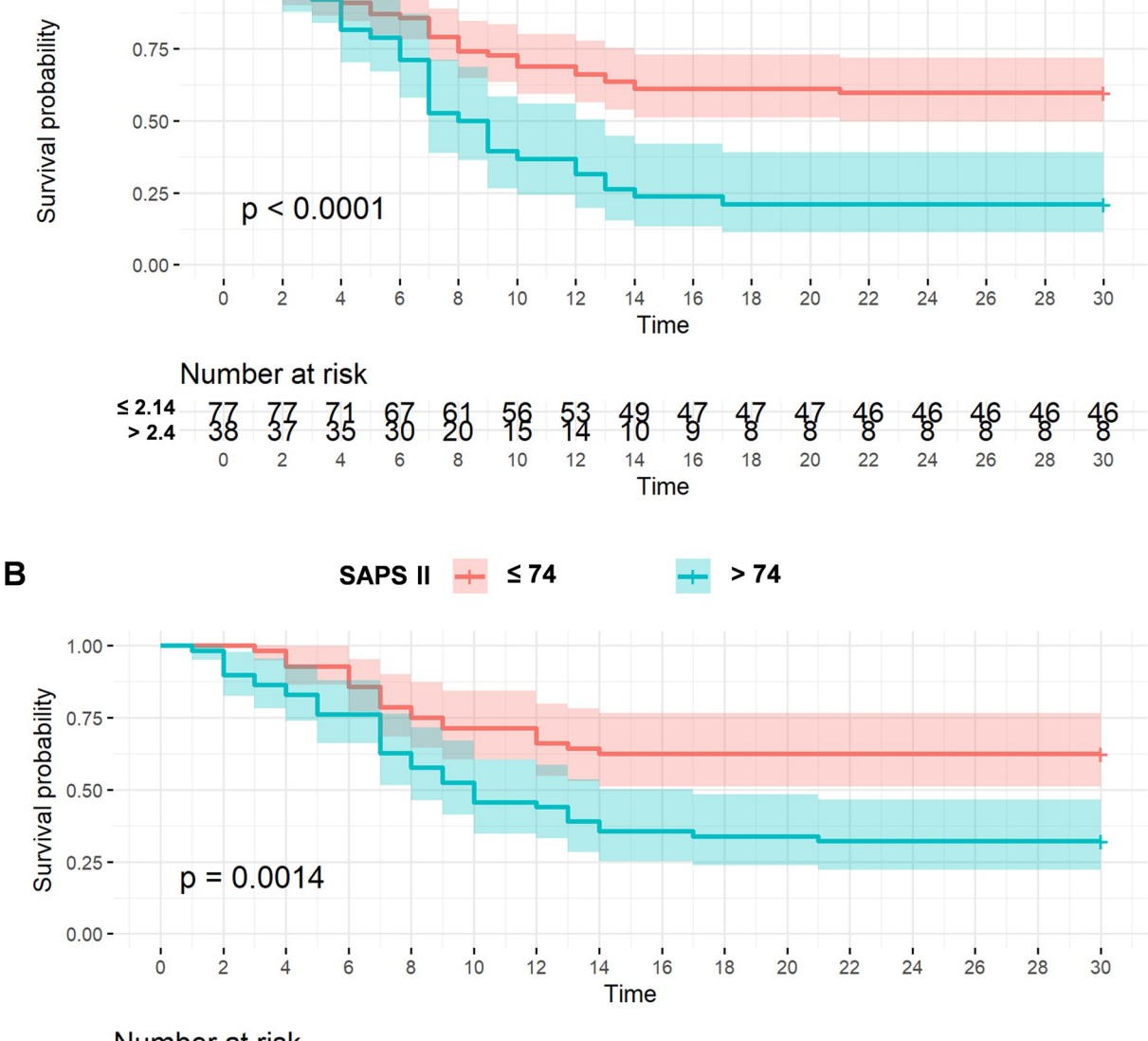

**Fig 4.** Kaplan-Meier curves for 30-days in-hospital survival according to the established (A) ΔPv-aCo₂/ΔCa-vO₂ ratio and (B) SAPS II cutoff values. A *p*-value of less than 0.05 was considered statistically significant. ΔPv-aCO₂/ΔCa-vO₂ ratio, central venous-to-arterial carbon dioxide difference combined with arterial-to-venous oxygen content difference; SAPS II, Simplified Acute Physiology Score II.

ScvO₂ translates the global cellular oxygenation status. ScvO₂ may be an indicator of mito-chondrial dysfunction where high ScvO₂ (≥ 80%) would reflect increased oxidative stress and decreased cellular respiration, while low ScvO₂ (<70%) would show less oxidative stress and increased cellular respiration [25]. In our study, there was no statistical difference in median ScvO₂ between survivors (75%) and non-survivors (76%), perhaps because the difference in

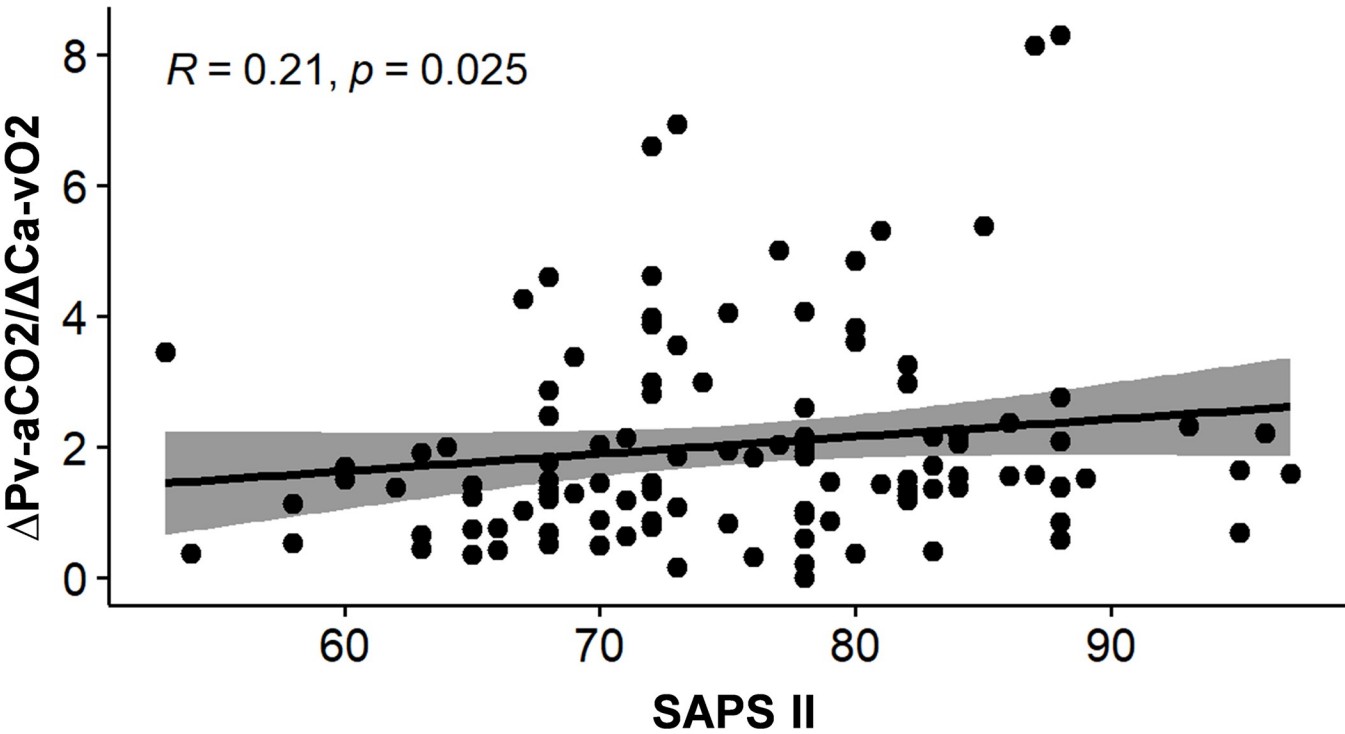

**Fig 5. Correlation of the ΔPv-aCo₂/ΔCa-vO₂ ratio with SAPS II in the patients with SARS-CoV-2-induced severe ARDS.** Spearman correlation coefficient at $p<0.05$ is shown. A $p$-value of less than 0.05 was considered statistically significant. ΔPv-aCO₂/ΔCa-vO₂ ratio, central venous-to-arterial carbon dioxide difference combined with arterial-to-venous oxygen content difference; SAPS II, Simplified Acute Physiology Score II.

survival appears with ScvO₂ <70% or >80% [26]. Moreover, variations in ScvO₂ may be caused not only by tissue hypoperfusion (decreased DO₂) but also by decreased arterial oxygen saturation, decreased hemoglobin, or increased VO₂ [27].

Although creatinine and base levels showed significant differences between the non-survivors and survivors ($p< 0.05$), only the base was statistically significant in the univariate Cox regression analysis. However, according to the multivariate Cox regression analysis, the base had no significant association with the non-survivors. For this reason, we do not consider that the renal part might also be involved in these mechanisms.

The relationship between carbon dioxide production (VCO₂) and CO is well documented, and values of ΔPv-aCO₂ >6mmHg suggest decreased tissue perfusion due to inappropriate blood flow or cardiac output. From the above, we understand that the increase in ΔPv-aCO₂ will be secondary to ischemic hypoxia [28]. In the multivariate Cox regression analysis for mortality, the ΔPv-aCO₂ had HR 0.97 (95% CI 0.9–1.04; $p = 0.484$) with the median for survivors of 5 mmHg and 6 mmHg for non-survivors ($p = 0.039$), which let us understand that in patients with severe ARDS secondary to SARS-CoV-2, their tissue hypoxia problems are to a lesser extent caused by ischemic hypoxia or circulatory flow disturbances.

The assessment of anaerobic metabolism may be confusing in the presence of alterations in arterial oxygen saturation, temperature, hemoglobin, and hydrogen ions which modify the CO₂ dissociation curve by changing the linear relationship between CO₂ content and pressure [28]. The increase in the ΔPv-aCO₂/ΔCa-vO₂ ratio is due to hypoxic hypoxia, being its leading cause of increased anaerobic metabolism. Its modification will be minimal due to ischemic hypoxia or circulatory blood flow alterations [22]. For such reason, in patients with severe ARDS secondary to SARS-CoV-2, we argue for higher anaerobic metabolism in the non-

survivors group (ΔPv-aCO$_2$/ΔCa-vO$_2$ ratio of 2.05) concerning survivors (ΔPv-aCO$_2$/ΔCa-vO$_2$ ratio of 1.35). Hypoxic hypoxia causes this increase in anaerobic metabolism secondary to severe hypoxemia characteristic of these patients.

There are contradictory results on ΔPv-aCO$_2$/ΔCa-vO$_2$ ratio and its relationship with mortality mainly because the cut-off point is poorly defined. The ranges oscillate between 1.4 to 1.7 mmHg/mL; values above this point are associated with increased mortality in different studies [13]. A recent meta-analysis indicates that ΔPv-aCO$_2$/ΔCa-vO$_2$ ratio predicts mortality in patients with septic shock, mainly when measured at 6 hours of admission (Risk Ratio (RR) = 1.89, 95% CI 1.48–2.41, $p$ = <0.01) but the best cut-off point is not defined [29]. In previous work, we documented that septic patients with ΔPv-aCO$_2$/ΔCa-vO$_2$ ratio >1.4 mmHg/mL, measured 24 hours after ICU admission, is related to an increased risk of death at 30 days (OR 5.49, 95% CI 1.07–28.09, $p$ = 0.04). Ninety-three percent of patients who did not survive had lactate >2 mmol/L [30]. Likewise, ScvO$_2$ ≥80% is related to a higher ΔPv-aCO$_2$/ΔCa-vO$_2$ ratio concerning patients with lower ScvO$_2$ [31]. The ΔPv-aCO$_2$/ΔCa-vO$_2$ ratio is superior to lactate in identifying anaerobic metabolism [3, 32]. We consider that lactate >2 mmol/L should be evaluated with ΔPv-aCO$_2$/ΔCa-vO$_2$ ratio; the increased latter suggests tissue hypoxia and increased anaerobic metabolism. Conversely, lactate levels >2 mmol/L without increased ΔPv-aCO$_2$/ΔCa-vO$_2$ ratio force us to reevaluate the origin of lactate [33].

The statistically significant variables in the multivariate Cox regression analysis for mortality in patients with severe ARDS secondary to SARS-CoV-2 were BMI, ΔPv-aCO$_2$/ΔCa-vO$_2$ ratio, and SAPS II. The cut-off limit for mortality for ΔPv-aCO$_2$/ΔCa-vO$_2$ ratio was >2.14 mmHg/mL and for SAPS II >74 points. What is important about those values is the lower number of variables used by the ΔPv-aCO$_2$/ΔCa-vO$_2$ ratio to SAPS II, making it an accessible and affordable prognostic tool.

The limitations of our study are the sample size (n = 115) and it is a single-center study. Although there is sufficient evidence regarding the usefulness of the ΔPv-aCO$_2$/ΔCa-vO$_2$ ratio to detect increased anaerobic metabolism, we do not have a "Gold Standard," which could be the respiratory quotient. Although the variable smoking between the two groups had a statistically significant difference, in the multivariate Cox regression analysis, this variable was not an independent risk factor for mortality of patients with acute respiratory distress syndrome related to COVID-19. However, smoking can increase CO$_2$ in blood gas analysis; thus, this issue could be another study limitation. Of the strengths, 100% of the patients were in ICU with IVM, a homogeneous population. In addition, the variables that modify the ΔPv-aCO$_2$/ΔCa-vO$_2$ ratio outside the context of increased anaerobic metabolism were not statistically significant. An important point is that the deterioration of lung function during SARS-CoV-2 infection induces alternative compensation mechanisms for oxygen uptake, such as the enhanced hemoglobin oxygen through a left shift of the oxygen dissociation curve, increasing perfusion, which modulates central venous blood gases [34]. Although the oxygen dissociation curve was not performed in this study, clinical relevance cannot be excluded. It needs further evaluation to determine their impact on the ΔPv-aCO$_2$/ΔCa-vO$_2$ ratio in the prognostic of COVID-19 patients.

The use of the ΔPv-aCO$_2$/ΔCa-vO$_2$ ratio as a predictor of mortality in patients with severe ARDS secondary to SARS-CoV-2 has not yet been proven. Finally, the gasometrical resource is affordable in most hospitals.

## Conclusion

In this study the ΔPv-aCO$_2$/ΔCa-vO$_2$ ratio >2.14 mmHg/mL was an independent risk factor for mortality (HR 1.17, 95% CI 1.06–1.29, $p$ = 0.001) in patients with severe ARDS secondary

to SARS-CoV-2. Hence, the $\Delta$Pv-a$CO_2$/$\Delta$Ca-v$O_2$ ratio could help determine the prognosis of these patients.

## Acknowledgments

The authors thank all the health workers, doctors, nurses, and support staff who worked in the UMAE, H. E. No. 14, Centro Médico Nacional "Adolfo Ruiz Cortines", IMSS Veracruz, who were fighting for COVID-19 together.

## Author Contributions

**Conceptualization:** Jesús Salvador Sánchez Díaz, Karla Gabriela Peniche Moguel, José Manuel Reyes-Ruiz, Orlando Rubén Pérez Nieto, Eder Iván Zamarrón López, María Verónica Calyeca Sánchez.

**Data curation:** Jesús Salvador Sánchez Díaz, Karla Gabriela Peniche Moguel, José Manuel Reyes-Ruiz.

**Formal analysis:** Jesús Salvador Sánchez Díaz, Karla Gabriela Peniche Moguel, José Manuel Reyes-Ruiz, Orlando Rubén Pérez Nieto.

**Funding acquisition:** Jesús Salvador Sánchez Díaz, José Manuel Reyes-Ruiz.

**Investigation:** Jesús Salvador Sánchez Díaz, Karla Gabriela Peniche Moguel, José Manuel Reyes-Ruiz, Orlando Rubén Pérez Nieto, Diego Escarramán Martínez, María Verónica Calyeca Sánchez.

**Methodology:** Jesús Salvador Sánchez Díaz, Karla Gabriela Peniche Moguel, José Manuel Reyes-Ruiz, Eder Iván Zamarrón López, María Verónica Calyeca Sánchez.

**Project administration:** Jesús Salvador Sánchez Díaz, Karla Gabriela Peniche Moguel.

**Resources:** Jesús Salvador Sánchez Díaz, Karla Gabriela Peniche Moguel, José Manuel Reyes-Ruiz.

**Software:** Jesús Salvador Sánchez Díaz, Karla Gabriela Peniche Moguel, José Manuel Reyes-Ruiz.

**Supervision:** Jesús Salvador Sánchez Díaz, Karla Gabriela Peniche Moguel, José Manuel Reyes-Ruiz.

**Validation:** Jesús Salvador Sánchez Díaz, Karla Gabriela Peniche Moguel, José Manuel Reyes-Ruiz, Orlando Rubén Pérez Nieto, Diego Escarramán Martínez, Eder Iván Zamarrón López, María Verónica Calyeca Sánchez.

**Visualization:** Jesús Salvador Sánchez Díaz, Karla Gabriela Peniche Moguel, José Manuel Reyes-Ruiz, Orlando Rubén Pérez Nieto, Diego Escarramán Martínez, Eder Iván Zamarrón López, María Verónica Calyeca Sánchez.

**Writing – original draft:** Jesús Salvador Sánchez Díaz, Karla Gabriela Peniche Moguel, José Manuel Reyes-Ruiz, Orlando Rubén Pérez Nieto, María Verónica Calyeca Sánchez.

**Writing – review & editing:** Jesús Salvador Sánchez Díaz, Karla Gabriela Peniche Moguel, José Manuel Reyes-Ruiz, Diego Escarramán Martínez, Eder Iván Zamarrón López.

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
