## [Decision Letter · Decision Letter 0]

3 Jan 2023

PONE-D-22-29034∆Pv-aCO2/∆Ca-vO2 ratio as predictor of mortality in patients with acute respiratory distress syndrome related to COVID-19PLOS ONE

Dear Dr. Reyes-Ruiz,

Thank you for submitting your manuscript to PLOS ONE. After careful consideration, we feel that it has merit but does not fully meet PLOS ONE’s publication criteria as it currently stands. Therefore, we invite you to submit a revised version of the manuscript that addresses the points raised during the review process.

We look forward to receiving your revised manuscript.

Kind regards,

Bora Çekmen

Academic Editor

PLOS ONE

Journal Requirements:

2. Please include your tables as part of your main manuscript and remove the individual files. Please note that supplementary tables (should remain/ be uploaded) as separate "supporting information" files"

Reviewers' comments:

Reviewer's Responses to Questions

**Comments to the Author**

1. Is the manuscript technically sound, and do the data support the conclusions?

Reviewer #1: Partly

Reviewer #2: Yes

2. Has the statistical analysis been performed appropriately and rigorously? 

Reviewer #1: Yes

Reviewer #2: Yes

3. Have the authors made all data underlying the findings in their manuscript fully available?

Reviewer #1: Yes

Reviewer #2: Yes

4. Is the manuscript presented in an intelligible fashion and written in standard English?

Reviewer #1: Yes

Reviewer #2: Yes

5. Review Comments to the Author

Reviewer #1: Thank you for the opportunity to review this manuscript which quantifies the effect of the ratio ΔPv-aCO2 / ΔCa-vO2 on the mortality of patients with ARDS secondary to SARS-CoV-2.

Major comment:

The methodology is sound. My main issue is the concluding statement : “Patients with ARDS secondary to SARS-CoV-2 have increased ΔPv-aCO2/ΔCa-vO2 ratio …” This phrasing implies a comparison of this ratio between patients w/ ARDS secondary to SARS-CoV-2 and patients without ARDS secondary to SARS-CoV-2. That is not the case, because this study presents data only from the first group. The conclusion should be modified to simply state something to the effect of: “In patients with ARDS secondary to SARS-CoV-2, non-survivors have increased ΔPv-aCO2/ΔCa-vO2 ratio compared to the survivors …” This modified phrasing more accurately circles back to the title and stated objective of the study, and it is supported by the data and methodology presented here.

Minor comments:

1) The 3rd paragraph of the Results section has this statement -- “Fig 3 and Fig 4A shows [sic] the Kaplan-Meier …” But Fig 3A is the ROC curve, while 4A is the Kaplan-Meier curve for ΔPv-aCO2/ΔCa-vO2.

2) I noticed a couple of grammatical errors “shows” (instead of “show”) – see [sic] above; and “shown” instead of “shows” (see end of 2nd paragraph of the Results section).

3) You use the two notations “Pv-aCO2” and “ΔPv-aCO2” interchangeably. If you mean for them to be interchangeable, then it would be better to just pick one and stay with it throughout the manuscript (to minimize confusion).

4) In the Introduction section, shouldn't there be a period between “... most commonly used cut-off point [8]” and “ScvO2 surrogates …” ?

5) In the Introduction, you wrote: “... but without inappropriate blood flow, CO2 will be poorly …” – this seems like an unintended double negative, so either remove "without" or change “inappropriate” to “appropriate”.

6) In the Introduction, what does the acronym “CR” stand for? It appears twice without definition.

7) The second to last sentence under the “Data Collection” paragraph – “Other variables ..” is incomplete.

Reviewer #2: It was a well-considered idea to research ARDS patients with hypoxemic hypoxia. ΔPvaCO2/ΔCa-vO2 ratio might be useful to detect the mortality and prognosis of severe Covid-19 patients. But there are some concerns about the article which I stated below.

1. In the Introduction section the sentences start with ‘The CR will increase by higher VCO2 or lower VO2…….’, you used CR for the abbreviation, but this abbreviation was not clearly defined before.

2. The authors stated that all the patient was intubated with this research. Was the PaO2/FiO2 ratio obtained before intubation or it came from after the intubation period?

3. Why did not the authors exclude patients with hematologic disease? Many diseases might affect the hemoglobin level. It would be better to state that in the Method section.

4. In table 1, smoking differed between the two groups, and it is well known that smoking can increase CO2 in blood gas analysis. Due to this issue, it would be better to add a limitation to this article. Also, it was shown that all the patients were Level 3 ARDS patients (With Berlin Criteria). So, it would be better to state that this research was held with severe ARDS patients in both title and method section.

5. Were all patients treated with the same vasopressors? If it is not, could you add this to the method section? Because vasopressor types or using vasopressors can affect the MV time and mortality in critically ill patients.

6. In the Conclusion section, there was a sentence starting with ‘This study is the first study regarding patients with ARDS secondary to SARS-CoV-2’. It would be better if you start the sentence with it has not shown yet instead of ‘this is the first study’.

6. PLOS authors have the option to publish the peer review history of their article (what does this mean?). If published, this will include your full peer review and any attached files.

Reviewer #1: No

Reviewer #2: No

---

## [Author Response · Author response to Decision Letter 0]

21 Jan 2023

January 20th, 2023

Emily Chenette

Editor-in-Chief

Bora Çekmen

Academic Editor

PlosONE

Dear Editor,

Enclosed, please find the revised version of the manuscript entitled: “∆Pv-aCO2/∆Ca-vO2 ratio as predictor of mortality in patients with acute respiratory distress syndrome related to COVID-19” with ID: PONE-D-22-29034, which we are submitting for consideration for publication in PLOS ONE.

We attended to all the suggestions of the reviewers and performed additional modifications to complete this work.

Journal Requirements:

Reply: Thank you very much for your comments. In this new submission, we ensured that our manuscript met PLOS ONE´s style requirements. The Postal Codes were removed from the affiliations, and the list of corresponding authors was modified, including the initials in parentheses after the email address.

2. Please include your tables as part of your main manuscript and remove the individual files. Please note that supplementary tables (should remain/ be uploaded) as separate "supporting information" files"

 Reply: Tables 1 and 2 were included as part of the main manuscript (Pages 18-20), and the individual files were removed.

Reply: References have been checked, and we have ensured they are complete and correct. In addition, we changed some of the references to more current ones, as shown below:

The reference: “12. Mallat J, Lemyze M, Tronchon L, Vallet B, Thevenin D. Use of venous-to-arterial carbon dioxide tension difference to guide resuscitation therapy in septic shock. World J Crit Care Med. 2016; 5(1):47-56. https://doi.org/10.5492/wjccm.v5.i1.47 PMID: 26855893” was changed to “12.Nassar B, Badr M, Van Grunderbeeck N, Temime J, Pepy F, Gasan Gaelle, et al. Central venous-to-arterial PCO2 difference as a marker to identify fluid responsiveness in septic shock. Sci Rep. 2021; 11(1):17256. https://doi.org/10.1038/s41598-021-96806-6 PMID: 34446823”

The reference: “15. Dubin A, Pozo MO, Hurtado J. Venoarterial PCO2-to-arteriovenous oxygen content difference ratio is a poor surrogate for anaerobic metabolism in hemodilution: an experimental study. Rev Bras Ter Intensiva. 2020; 32(1):115-122. https://doi.org/10.5935/0103-507x.20200017 PMID: 32401981” was changed to “15. Dubin A, Ferrara G, Edul VSK, Martins E, Canales HS, Canullán C, et al. Venoarterial PCO2-to-arteriovenous oxygen content difference ratio is a poor surrogate for anaerobic metabolism in hemodilution: an experimental study. Ann Intensive Care. 2017; 7(1):65. https://doi.org/10.1186/s13613-017-0288-z PMID: 28608134”

The reference: “16. Zubieta-Calleja G, Zubieta-DeUrioste N. Pneumolysis and "Silent Hypoxemia" in COVID-19. Indian J Clin Biochem. 2020 Nov 9;36(1):1-5. doi: 10.1007/s12291-020-00935-0” was changed to “16. Al-Kuraishy HM, Al-Gareeb A, Al-Hamash SM, Cavalu S, El-Bouseary M, Sonbol F, et al. Changes in the Blood Viscosity in Patients With SARS-CoV-2 Infection. Front Med. 2022;17;9: 876017. https://doi.org/10.3389/fmed.2022.876017 PMID: 35783600”

The reference: “17. Arora S, Tantia P. Physiology of Oxygen Transport and its Determinants in Intensive Care Unit. Indian J Crit Care Med. 2019 Sep;23(Suppl 3): S172-S177. DOI: 10.5005/jp-journals-10071-23246” was changed to “17. Bracegirdle L, Jackson A, Beecham R, Burova M, Hunter E, Hamilton LG, et al. Dynamic blood oxygen indices in mechanically ventilated COVID-19 patients with acute hypoxic respiratory failure: A cohort study. PLoS One. 2022;17(6): e0269471. https://doi.org/10.1371/journal.pone.0269471 PMID: 35687543”

The reference: “20. Kattan E, Hernández G. The role of peripheral perfusion markers and lactate in septic shock resuscitation. Journal of Intensive Medicine, https://doi.org/10.1016/j.jointm.2021.11.002” was changed to “20. Cruces P, Retamal J, Hurtado DE, Erranz B, Iturrieta P, González C, et al. A physiological approach to understand the role of respiratory effort in the progression of lung injury in SARS-CoV-2 infection. Crit Care. 2020;24(1): 494. https://doi.org/10.1186/s13054-020-03197-7 PMID: 32778136”

The reference: “32. Hassanein A, Abbas I, Mohammed R. Central blood gases versus lactate level for assessment of initial resuscitation success in patients with sepsis in critical care, Egyptian Journal of Anaesthesia. 2022. 38:1, 439-445, DOI: 10.1080/11101849.2022.2108196” was changed to “32. McDonald CI, Brodie D, Schmidt M, Hay K, Shekar K. Elevated Venous to Arterial Carbon Dioxide Gap and Anion Gap Are Associated with Poor Outcome in Cardiogenic Shock Requiring Extracorporeal Membrane Oxygenation Support. ASAIO J. 2021;67(3): 263-269. https://doi.org/10.1097/MAT.0000000000001215 PMID: 33627599”

Reviewers' comments: Reviewer's Responses to Questions

Reply: Thank you very much for your comments.

Comments to the Author

1. Is the manuscript technically sound, and do the data support the conclusions?

Reviewer #1: Partly

Reviewer #2: Yes

2. Has the statistical analysis been performed appropriately and rigorously? 

Reviewer #1: Yes

Reviewer #2: Yes

3. Have the authors made all data underlying the findings in their manuscript fully available?

Reviewer #1: Yes

Reviewer #2: Yes

4. Is the manuscript presented in an intelligible fashion and written in standard English?

Reviewer #1: Yes

Reviewer #2: Yes

5. Review Comments to the Author

Reviewer #1: Thank you for the opportunity to review this manuscript which quantifies the effect of the ratio ΔPv-aCO2 / ΔCa-vO2 on the mortality of patients with ARDS secondary to SARS-CoV-2. 

Major comment: 

The methodology is sound. My main issue is the concluding statement : “Patients with ARDS secondary to SARS-CoV-2 have increased ΔPv-aCO2/ΔCa-vO2 ratio …” This phrasing implies a comparison of this ratio between patients w/ ARDS secondary to SARS-CoV-2 and patients without ARDS secondary to SARS-CoV-2. That is not the case, because this study presents data only from the first group. The conclusion should be modified to simply state something to the effect of: “In patients with ARDS secondary to SARS-CoV-2, non-survivors have increased ΔPv-aCO2/ΔCa-vO2 ratio compared to the survivors …” This modified phrasing more accurately circles back to the title and stated objective of the study, and it is supported by the data and methodology presented here.

Reply: Thank you very much for your comments. We modified the concluding statement according to the reviewer's comment to refer more precisely to the title and stated objective of our study, which is supported by the data and methodology presented here.

The concluding statement: “Patients with ARDS secondary to SARS-CoV-2 have increased ∆Pv-aCO2/∆Ca-vO2 ratio due to hypoxia secondary to severe hypoxemia…” was changed to “In patients with ARDS secondary to SARS-CoV-2, non-survivors have an increased ∆Pv-aCO2/∆Ca-vO2 ratio due to hypoxia secondary to severe hypoxemia…”

Minor comments: 

1) The 3rd paragraph of the Results section has this statement -- “Fig 3 and Fig 4A shows [sic] the Kaplan-Meier …” But Fig 3A is the ROC curve, while 4A is the Kaplan-Meier curve for ΔPv-aCO2/ΔCa-vO2.

Reply: We agree with the reviewer that Figure 3 corresponds to the ROC curve, therefore in this new submission, this error was corrected as follows:

“The AUC for ∆Pv-aCO2/∆Ca-vO2 ratio was 0.691(95% CI 0.598-0.774, p= 0.0001), with a best cut-off point of >2.14 mmHg/mL (sensitivity 49.18%, specificity 85.19%, positive likelihood ratio (LR+) 3.32, and negative likelihood ratio (LR-) 0.6) (Fig 3A). The best cut-off point obtained by Youden's index for SAPS II was >74 points (AUC= 0.696 (95% CI 0.603-0.778, p= 0.0001), sensitivity 65.57%, specificity 64.81%, LR+ 1.86, and LR- 0.53) (Fig 3B). Fig 4A shows the Kaplan-Meier curve of the ∆Pv-aCO2/∆Ca-vO2 ratio for 30-day survival, showing a statistically significant difference between survivors and non-survivors with a cut-off point of >2.14 mmHg/mL.”

2) I noticed a couple of grammatical errors “shows” (instead of “show”) – see [sic] above; and “shown” instead of “shows” (see end of 2nd paragraph of the Results section).

Reply: The words “shows” and “shown” were corrected according to the reviewer's comment.

“Table 2 show the Cox regression analysis for mortality in patients with ARDS secondary to SARS-CoV-2. In the univariate analysis, the variables BMI, smoking, Diabetes, SAPS II, vasopressor, pH, base, ∆Pv-aCO2, and ∆Pv-aCO2/∆Ca-vO2 ratio had statistical significance. In the multivariable analysis, BMI (HR 1.04, 95% CI 1.01-1.08, p= 0.007), SAPS II (HR 1.04, 95% CI 1.01-1.08, p= 0.005), and ∆Pv-aCO2/∆Ca-vO2 ratio (HR 1.17, 95% CI 1.06-1.29, p= 0.001) maintained statistical significance. Fig 2 shows the final Cox regression model for mortality in patients with COVID-19.”

3) You use the two notations “Pv-aCO2” and “ΔPv-aCO2” interchangeably. If you mean for them to be interchangeable, then it would be better to just pick one and stay with it throughout the manuscript (to minimize confusion).

Reply: In this new submission, the notation “Pv-aCO2” was changed to “ΔPv-aCO2,” and this word was maintained throughout the manuscript to minimize confusion, as suggested by the reviewer.

“ΔPv-aCO2/ΔCa−vO2 ratio= ΔPv-aCO2/ΔCa−vO2” (Definitions, Section of Material and Methods).

“The correlation between ΔPv-aCO2/ΔCa-vO2 and SAPS II was calculated using the Spearman correlation test. All analyses were performed using R Studio v4.03 statistical (R Foundation, Vienna, Austria) and SPSS v.25 software (IBM, New York, USA)” (Section of Statistical analysis).

In addition, an error in Figure 2 was detected (“Pa-vCO2” instead of “ΔPv-aCO2”). Therefore the figure was modified and uploaded in this submission of the manuscript.

4) In the Introduction section, shouldn't there be a period between “... most commonly used cut-off point [8]” and “ScvO2 surrogates …” ?

Reply: We added a period between “... most commonly used cut-off point [8]” and “ScvO2 surrogates …” as suggested by the reviewer.

“… most commonly used cut-off point [8]. ScvO2 surrogates…”

5) In the Introduction, you wrote: “... but without inappropriate blood flow, CO2 will be poorly …” – this seems like an unintended double negative, so either remove "without" or change “inappropriate” to “appropriate”.

Reply: In this new submission, the sentence in the Introduction was modified by changing “inappropriate” to “appropriate” according to the reviewer's commentary:

“… but without appropriate blood flow, CO2 will be poorly …”

6) In the Introduction, what does the acronym “CR” stand for? It appears twice without definition.

Reply: The acronym “CR” is a typo. In this sense, the correct acronym is “RQ”. This word was corrected in this new submission.

7) The second to last sentence under the “Data Collection” paragraph – “Other variables ..” is incomplete.

Reply: In this new submission, the second to last sentence under the “Data Collection” paragraph – “Other variables ..” was completed as suggested by the reviewer.

“Other variables such as creatinine, D-dimer, C-reactive protein, fibrinogen, glutamic oxaloacetic transaminase [GOT], glutamic pyruvic transaminase [GPT], and vasopressor were also included in this study.”

Reviewer #2: It was a well-considered idea to research ARDS patients with hypoxemic hypoxia. ΔPvaCO2/ΔCa-vO2 ratio might be useful to detect the mortality and prognosis of severe Covid-19 patients. But there are some concerns about the article which I stated below.

1. In the Introduction section the sentences start with ‘The CR will increase by higher VCO2 or lower VO2…….’, you used CR for the abbreviation, but this abbreviation was not clearly defined before.

Reply: Thank you very much for your comments. The acronym “CR” is a typo, the correct acronym is “RQ”. This word was corrected in this new submission as follows:

‘The RQ will increase by higher VCO2 or lower VO2…….’

2. The authors stated that all the patient was intubated with this research. Was the PaO2/FiO2 ratio obtained before intubation or it came from after the intubation period?

Reply: The arterial and central venous blood gases, including PaO2/FiO2 ratio, were obtained 30 minutes after intubation and processed in the GEM® PREMIER™4000 with iQM® equipment. This sentence was included in this new submission (Section of Material and methods, Study design and patients).

3. Why did not the authors exclude patients with hematologic disease? Many diseases might affect the hemoglobin level. It would be better to state that in the Method section.

Reply: We agree with the reviewer's comment. Because many diseases might affect the hemoglobin level as hematologic diseases, we decided to exclude patients with this characteristic from the beginning of this study. This exclusion criterion was included in this submission as follows:

“(1) Patients with diseases that could affect the hemoglobin level as hematologic diseases, (2) patients with an incomplete variable registry, or (3) pregnant were excluded from this study.” (Section of Material and methods, Study design and patients).

4. In table 1, smoking differed between the two groups, and it is well known that smoking can increase CO2 in blood gas analysis. Due to this issue, it would be better to add a limitation to this article. Also, it was shown that all the patients were Level 3 ARDS patients (With Berlin Criteria). So, it would be better to state that this research was held with severe ARDS patients in both title and method section.

Reply: Although the variable “smoking” between the two groups had a statistically significant difference, in the multivariable COX regression analysis, this variable was not found to be an independent risk factor for mortality of patients with acute respiratory distress syndrome related to COVID-19. However, we agree with the reviewer that smoking can increase CO2 in blood gas analysis. Due to this issue, we decided to add a limitation to this study in the Discussion section as follows: 

“Although the variable “smoking” between the two groups had a statistically significant difference, in the multivariable COX regression analysis, this variable was not found to be an independent risk factor for mortality of patients with acute respiratory distress syndrome related to COVID-19. However, smoking can increase CO2 in blood gas analysis; thus, this issue could be another study limitation.”

On the other hand, we agree with the reviewer that this study was conducted in patients with severe ARDS; therefore, this issue was indicated in the title as well as in the method section: 

The title was changed to “The ∆Pv-aCO2/∆Ca-vO2 ratio as a predictor of mortality in patients with severe acute respiratory distress syndrome related to COVID-19”.

“Inclusion criteria were: (1) age >18 years, (2) any gender, (3) confirmed SARS-CoV-2 infection by reverse transcriptase polymerase chain reaction (RT-PCR), and (4) severe ARDS (PaO2/FiO2 ≤100 mmHg) defined according to Berlin criteria [18] with invasive mechanical ventilation (IVM).” (Section of Material and methods, Study design and patients).

5. Were all patients treated with the same vasopressors? If it is not, could you add this to the method section? Because vasopressor types or using vasopressors can affect the MV time and mortality in critically ill patients.

Reply: All patients were treated with norepinephrine as the only vasopressor. This sentence was included in the Material and Methods Section in this new submission:

“All patients were intubated in the ICU and treated with norepinephrine as only vasopressor”.

6. In the Conclusion section, there was a sentence starting with ‘This study is the first study regarding patients with ARDS secondary to SARS-CoV-2’. It would be better if you start the sentence with it has not shown yet instead of ‘this is the first study’.

Reply: The sentence starting with “This study is the first study regarding patients with ARDS secondary to SARS-CoV-2…” was changed to “The use of the ∆Pv-aCO2/∆Ca-vO2 ratio as a predictor of mortality in patients with severe ARDS secondary to SARS-CoV-2 has not yet been proven.” (Conclusion Section).

6. PLOS authors have the option to publish the peer review history of their article (what does this mean?). If published, this will include your full peer review and any attached files.

Do you want your identity to be public for this peer review? For information about this choice, including consent withdrawal, please see our Privacy Policy.

Reviewer #1: No

Reviewer #2: No

Please let me know about above questions.

By signing this letter, we acknowledge that all the authors participated sufficiently to take public responsibility for its content. All of the authors have given their consent for submission to the journal. Further, we have no commercial associations which impact this work.

Best regards, 

Professor José Manuel Reyes 

Mexican Social Security Institute (IMSS)

E-mail: jose.reyesr@imss.gob.mx

---

## [Editor Report · Decision Letter 1]

27 Feb 2023

PONE-D-22-29034R1The ∆Pv-aCO2/∆Ca-vO2 ratio as a predictor of mortality in patients with severe acute respiratory distress syndrome related to COVID-19PLOS ONE

Dear Dr. Reyes-Ruiz,

Thank you for submitting your manuscript to PLOS ONE. After careful consideration, we feel that it has merit but does not fully meet PLOS ONE’s publication criteria as it currently stands. Therefore, we invite you to submit a revised version of the manuscript that addresses the points raised during the review process.

ACADEMIC EDITOR:1. There is a difference between the abstract and the main article with the changes you made in the conclusion part. This needs to be fixed again.

2. In the Abstract section, a few sentences of explanation are required about which parameters you are examining in the Method section. You must do it in a way that complies with the word count limit.

3. You should change the changes you made in the results section of the main text in the abstract section as well.

4. In the Abstract section, “Results” should be written instead of “Data synthesis”.

We look forward to receiving your revised manuscript.

Kind regards,

Bora Çekmen

Academic Editor

PLOS ONE

Journal Requirements:

Additional Editor Comments (if provided):

Dear Author,

Thank you for making the necessary changes in line with the reviewers' suggestions. Some additional technical fixes are required. These will be reviewed after they are made.

Best Regards

---

## [Author Response · Author response to Decision Letter 1]

7 Mar 2023

March 07th, 2023

Emily Chenette

Editor-in-Chief

Bora Çekmen

Academic Editor

PlosONE

Dear Editor,

Enclosed, please find the revised version of the manuscript entitled: “∆Pv-aCO2/∆Ca-vO2 ratio as predictor of mortality in patients with acute respiratory distress syndrome related to COVID-19” with ID: PONE-D-22-29034, which we are submitting for consideration for publication in PLOS ONE.

We attended to all the suggestions of the ACADEMIC EDITOR and performed additional modifications to complete this work.

RESPONSE TO ACADEMIC EDITOR' COMMENTS

ACADEMIC EDITOR:

1. There is a difference between the abstract and the main article with the changes you made in the conclusion part. This needs to be fixed again.

Reply: Thank you very much for your comments. In this new submission, to eliminate the differences between the abstract and the main article with the changes previously made in the conclusion part, the conclusion was modified in the Abstract and Conclusion sections, as follows:

In the Abstract:

The sentence: Conclusions: “Patients with severe ARDS secondary to SARS-CoV-2 have increased ∆Pv-aCO2/∆Ca-vO2 ratio due to severe hypoxemia. Hence, ∆Pv-aCO2/∆Ca-vO2 ratio could be used as a predictor of COVID-19 mortality” was changed to “The ∆Pv-aCO2/∆Ca-vO2 ratio could be used as a predictor of mortality in patients with severe ARDS secondary to SARS-CoV-2.”

In the Conclusions

The sentence: “In patients with severe ARDS secondary to SARS-CoV-2, non-survivors have an increased ∆Pv-aCO2/∆Ca-vO2 ratio due to hypoxia secondary to severe hypoxemia, reflecting increased anaerobic metabolism. In contrast, circulatory blood flow alterations secondary to ischemic hypoxia are less relevant. The ∆Pv-aCO2/∆Ca-vO2 ratio >2.14 mmHg/mL is an independent risk factor for mortality (HR 1.17, 95% CI 1.06-1.29, p= 0.001) in this patient population. The ∆Pv-aCO2/∆Ca-vO2 ratio could help determine the prognosis” was changed to “The ∆Pv-aCO2/∆Ca-vO2 ratio >2.14 mmHg/mL is an independent risk factor for mortality (HR 1.17, 95% CI 1.06-1.29, p= 0.001) in patients with severe ARDS secondary to SARS-CoV-2. Hence, the ∆Pv-aCO2/∆Ca-vO2 ratio could help determine the prognosis of these patients.”

2. In the Abstract section, a few sentences of explanation are required about which parameters you are examining in the Method section. You must do it in a way that complies with the word count limit.

Reply: In this new submission, the abstract was modified complying with the word count limit to include a few sentences explaining the parameters that were examined in the Material and Methods section (number of words: 273 words):

“Abstract

Objective: To evaluate the central venous-to-arterial carbon dioxide difference combined with arterial-to-venous oxygen content difference (∆Pv-aCO2/∆Ca-vO2 ratio) as a predictor of mortality in patients with COVID-19-related severe acute respiratory distress syndrome (ARDS). 

Methods: Patients admitted to the intensive care unit with severe ARDS secondary to SARS-CoV-2, and invasive mechanical ventilation were included in this single-center and retrospective cohort study performed between April 18, 2020, and January 18, 2022. The tissue perfusion indexes (lactate, central venous oxygen saturation [ScvO2], and venous-to-arterial carbon dioxide pressure difference [∆Pv-aCO2]), anaerobic metabolism index (∆Pv-aCO2/∆Ca-vO2 ratio), and severity index (Simplified Acute Physiology Score II [SAPSII]) were evaluated to determine its association with the mortality through Cox regression analysis, Kaplan-Meier curve and receiver operating characteristic (ROC) curve.

Results: One hundred fifteen patients were included in the study and classified into two groups, the survivor group (n= 54) and the non-survivor group (n= 61). The lactate, ScvO2, ∆Pv-aCO2, and ∆Pv-aCO2/∆Ca-vO2 ratio medians were 1.6 mEq/L, 75%, 5 mmHg, and 1.56 mmHg/mL, respectively. The ∆Pv-aCO2/∆Ca-vO2 ratio (Hazard Ratio (HR)= 1.17, 95% confidence interval (CI)= 1.06-1.29, p= 0.001) was identified as a mortality biomarker for patients with COVID-19-related severe ARDS. The area under the curve for ∆Pv-aCO2/∆Ca-vO2 ratio was 0.691 (95% CI 0.598-0.774, p= 0.0001). The best cut-off point for ∆Pv-aCO2/∆Ca-vO2 ratio was >2.14 mmHg/mL, with a sensitivity of 49.18%, specificity of 85.19%, a positive likelihood of 3.32, and a negative likelihood of 0.6. The Kaplan-Meier curve showed that survival rates were significantly worse in patients with values greater than this cut-off point.

Conclusions: The ∆Pv-aCO2/∆Ca-vO2 ratio could be used as a predictor of mortality in patients with severe ARDS secondary to SARS-CoV-2.”

3. You should change the changes you made in the results section of the main text in the abstract section as well.

Reply: The changes that were made in the results section of the main text are mentioned in the Abstract Section of this resubmission of the manuscript.

4. In the Abstract section, “Results” should be written instead of “Data synthesis”.

Reply: In the Abstract section, “Data synthesis” was changed to “Results”.

Thank you very much for your comments. Please let me know about above questions.

By signing this letter, we acknowledge that all the authors participated sufficiently to take public responsibility for its content. All of the authors have given their consent for submission to the journal. Further, we have no commercial associations which impact this work.

Best regards, 

Professor José Manuel Reyes 

Mexican Social Security Institute (IMSS)

E-mail: jose.reyesr@imss.gob.mx

---

## [Decision Letter · Decision Letter 2]

13 Jun 2023

PONE-D-22-29034R2The ∆Pv-aCO2/∆Ca-vO2 ratio as a predictor of mortality in patients with severe acute respiratory distress syndrome related to COVID-19PLOS ONE

Dear Dr. Reyes-Ruiz,

Thank you for submitting your manuscript to PLOS ONE. After careful consideration, we feel that it has merit but does not fully meet PLOS ONE’s publication criteria as it currently stands. Therefore, we invite you to submit a revised version of the manuscript that addresses the points raised during the review process.

ACADEMIC EDITOR: The paper is indeed interesting and deserves publication; the points highlighted by the reviewer #3 must be resolved before we can move forward.

We look forward to receiving your revised manuscript.

Kind regards,

Samuele Ceruti

Academic Editor

PLOS ONE

Journal Requirements:

Reviewers' comments:

Reviewer's Responses to Questions

**Comments to the Author**

1. If the authors have adequately addressed your comments raised in a previous round of review and you feel that this manuscript is now acceptable for publication, you may indicate that here to bypass the “Comments to the Author” section, enter your conflict of interest statement in the “Confidential to Editor” section, and submit your "Accept" recommendation.

Reviewer #3: All comments have been addressed

Reviewer #4: All comments have been addressed

2. Is the manuscript technically sound, and do the data support the conclusions?

Reviewer #3: Yes

Reviewer #4: Yes

3. Has the statistical analysis been performed appropriately and rigorously? 

Reviewer #3: Yes

Reviewer #4: Yes

4. Have the authors made all data underlying the findings in their manuscript fully available?

Reviewer #3: Yes

Reviewer #4: Yes

5. Is the manuscript presented in an intelligible fashion and written in standard English?

Reviewer #3: Yes

Reviewer #4: Yes

6. Review Comments to the Author

Reviewer #3: I thank the editor for sending me this manuscript for review and revision.

Interesting work with some points to be clarified, with the methods poorly expressed and explicated making the work difficult to reproduce and unclear in its development. The authors should be clearer on several points that, thoroughly explored would make the manuscript clear and comprehensive, without gray spots that could be misinterpreted.

In any case, an interesting work to be reworked further but its results are not negligible.

Materials and Methods.

I would have some questions for the authors with respect to the inclusion and exclusion criteria and also for some subsequent points.

1. The authors state that patients with diseases that could affect hemoglobin level were excluded. But how come only those? As we see later in the formulas, diseases that can affect CO2 should also have been excluded, since they also lead to situations where CO2 is compensated by an adaptation over time in which the patient lives normally but with CO2 that could be "corrected."

2. A further claim made, later by the authors is that all patients were intubated in the ICU and treated with norepinephrine as the only vasopressor.

This statement assumes that all patients treated by the authors and examined did not need any other inotropic support. So this was an inclusion or exclusion criterion, depending on how we see the reasoning. Correct? All patients had adequate contractile function and never needed any inotropic support other than vasoconstrictor norepinephrine?

3. The authors state that an on-call critical care specialist scheduled the IVM, does this mean everything and does it mean nothing, how was the IVM "scheduled," were ARDS ventilation criteria met, was advanced monitoring (esophageal pressure, transpulmonary) used for monitoring, or is this not a procedure that usual in the ICU in question? Was ventilation guided by driving pressure or solely by plateau pressure? These are all details that are helpful in understanding, later also, the patient's aerobic and anaerobic consumption and assessment.

Other points, which I did not find in the methods, are both the sedation of the patients, how were they managed? And also the ventilatory modalities? When were these variables assessed? In the hyperacute phase or retrospectively? These points are important

4. Regarding the first point, it is important because depending on the level of sedation that is applied to the patient, the patient has more or less consumption than he needs. If we sedate or even curarize a patient we know that we conspicuously lower the patient's consumption and requirement, on the other hand a patient who is breathing independently, perhaps in "conflict" with the ventilator, or with a fever or ongoing sepsis needs more requirement because he or she has a higher "consumption."

5. This also affects the ventilatory modalities adopted during the study and for the evaluation period, perhaps in the very early phase there is not this big consumption since the patient is put completely at "rest," conversely, during awakening and weaning we need more requirements. All these points should be considered by the authors always in this methods section, or at least adequately described.

Discussion

6. An interesting part that the authors could also analyze is the presence of significance between live and dead with respect to creatinine and bases. Which is well seen in the comparison between the two groups, then in the Cox Regression the creatinine does not become significant but the bases remain significant. Did the authors think that the renal part might also be involved in these mechanisms?

7. One thing that has not been taken into consideration by the authors is also the fact that patients are affected by hypoxia, and thus come to the hospital, after a period when the hemoglobin dissociation curve (p50) has shifted, allowing patients to better "tolerate" hypoxia as shown by many present studies. This aspect could also influence the outcome of the work. What do the authors think about this aspect? Did they take it into consideration?

Conclusion

8. I would be slightly softer, not stating that the ratio is an independent risk factor, but stating that in this study the ratio was found to be an independent risk factor, precisely also in light of the limitations of the study.

Reviewer #4: (No Response)

7. PLOS authors have the option to publish the peer review history of their article (what does this mean?). If published, this will include your full peer review and any attached files.

Reviewer #3: No

Reviewer #4: **Yes: **MUMOLI NICOLA, MD - MAGENTA HOSPITAL, MAGENTA, ITALY

---

## [Author Response · Author response to Decision Letter 2]

15 Jul 2023

July 13th, 2023

Emily Chenette

Editor-in-Chief

Bora Çekmen

Academic Editor

PlosONE

Dear Editor,

Enclosed, please find the revised version of the manuscript entitled: “The ∆Pv-aCO2/∆Ca-vO2 ratio as a predictor of mortality in patients with severe acute respiratory distress syndrome related to COVID-19” [PONE-D-22-29034R2] - [EMID:325f746b4f6fc97d], which we are submitting for consideration for publication in PLOS ONE.

We attended to all the suggestions of the Reviewer #3 and performed additional modifications to complete this work.

RESPONSE TO REVIEWER'S COMMENTS

Reviewer #3: I thank the editor for sending me this manuscript for review and revision.

Interesting work with some points to be clarified, with the methods poorly expressed and explicated making the work difficult to reproduce and unclear in its development. The authors should be clearer on several points that, thoroughly explored would make the manuscript clear and comprehensive, without gray spots that could be misinterpreted.

Reply: Thank you very much for your review and comments. We have made the necessary changes and modifications to the manuscript.

In any case, an interesting work to be reworked further but its results are not negligible.

Materials and Methods.

I would have some questions for the authors with respect to the inclusion and exclusion criteria and also for some subsequent points.

1. The authors state that patients with diseases that could affect hemoglobin level were excluded. But how come only those? As we see later in the formulas, diseases that can affect CO2 should also have been excluded, since they also lead to situations where CO2 is compensated by an adaptation over time in which the patient lives normally but with CO2 that could be "corrected."

Reply: We agree with the reviewer. Although not previously mentioned, we also excluded patients with chronic obstructive pulmonary disease (COPD), known neuromuscular disease or known hyperbaric respiratory failure which could affect CO2 levels. We include these exclusion criteria in this new submission as follows (Page 4, Material and Methods Section): 

(1) Patients with diseases that could affect the hemoglobin or CO2 levels as hematologic diseases, chronic obstructive pulmonary disease (COPD), known neuromuscular disease or known hyperbaric respiratory failure; (2) patients with an incomplete variable registry; or (3) pregnant were excluded from this study.

2. A further claim made, later by the authors is that all patients were intubated in the ICU and treated with norepinephrine as the only vasopressor.

This statement assumes that all patients treated by the authors and examined did not need any other inotropic support. So this was an inclusion or exclusion criterion, depending on how we see the reasoning. Correct? All patients had adequate contractile function and never needed any inotropic support other than vasoconstrictor norepinephrine?

Reply: The sentence “All patients were intubated in the ICU and an treated with norepinephrine as the only vasopressor” is a typing mistake. In our study, all patients were intubated in the ICU but some of them received norepinephrine (n= 26, 22.60%) as the only vasopressor (Table 1). These patients did not require ionotropic support other than the vasoconstrictor norepinephrine. Thus, inotropic support was not an inclusion or exclusion criterion.

In this new submission (Page 4, Material and Methods Section), the sentence: “All patients were intubated in the ICU and an treated with norepinephrine as the only vasopressor” was changed to “All patients were intubated in the ICU and some of them received norepinephrine (n= 26, 22.60%) as the only vasopressor. These patients did not require ionotropic support other than the vasoconstrictor norepinephrine.”

3. The authors state that an on-call critical care specialist scheduled the IVM, does this mean everything and does it mean nothing, how was the IVM "scheduled," were ARDS ventilation criteria met, was advanced monitoring (esophageal pressure, transpulmonary) used for monitoring, or is this not a procedure that usual in the ICU in question? Was ventilation guided by driving pressure or solely by plateau pressure? These are all details that are helpful in understanding, later also, the patient's aerobic and anaerobic consumption and assessment.

Other points, which I did not find in the methods, are both the sedation of the patients, how were they managed? And also the ventilatory modalities? When were these variables assessed? In the hyperacute phase or retrospectively? These points are important

Reply: We agree with the reviewer that these points are important in order to provide further information. First, lung-protective mechanical ventilation was applied in the volume assist-controlled mode using the Puritan Bennett 840 ventilator (Medtronic; Carlsbad, California, USA), with the following settings: tidal volume of 6 mL/Kg predicted body wight, plateau pressure ≤27 cmH2O, and driving pressure ≤15 cmH2O.

Second, the patients were sedated using propofol and mechanical ventilation was started. After 30 min of ventilation in a supine positioning the ventilatory variables, including perfusion indexes and anaerobic metabolism, were assessed.

In this new submission, we included these points in the Material and methods Section (Pages 4-5) as follows:

“All the patients were sedated using propofol and mechanical ventilation was started. Lung-protective mechanical ventilation was applied in the volume assist-controlled mode using the Puritan Bennett 840 ventilator (Medtronic; Carlsbad, California, USA), with the following settings: tidal volume of 6 mL/Kg predicted body wight, plateau pressure ≤27 cmH2O, and driving pressure ≤15 cmH2O. After 30 min of ventilation in a supine positioning the ventilatory variables, including perfusion indexes and anaerobic metabolism, were assessed.”

4. Regarding the first point, it is important because depending on the level of sedation that is applied to the patient, the patient has more or less consumption than he needs. If we sedate or even curarize a patient we know that we conspicuously lower the patient's consumption and requirement, on the other hand a patient who is breathing independently, perhaps in "conflict" with the ventilator, or with a fever or ongoing sepsis needs more requirement because he or she has a higher "consumption."

Reply: The propofol infusion was administered to maintain a Richmond Agitation-Sedation Scale (RASS) score of -3 (moderate sedation; patient had any movement in response to voice, but no eye contact) and overcome ventilator asynchrony, obtain a level of awake sedation, optimizing the patient´s respiratory status without effects on respiratory pattern, respiratory drive, and arterial and central venous blood gases.

This response was included in the Material and methods Section (Page 5, paragraph 2) of this new submission.

5. This also affects the ventilatory modalities adopted during the study and for the evaluation period, perhaps in the very early phase there is not this big consumption since the patient is put completely at "rest," conversely, during awakening and weaning we need more requirements. All these points should be considered by the authors always in this methods section, or at least adequately described.

Reply: We agree with the reviewer. The level of sedation that is applied to the patient could affect the ventilatory variables adopted during the evaluation study. However, this evaluation was performed in the early phase (after 30 min of ventilation in a supine positioning) where there is not a big consumption since the patient is put completely at “rest”. This point was answered as follows:

“All the patients were sedated using propofol and mechanical ventilation was started. Lung-protective mechanical ventilation was applied in the volume assist-controlled mode using the Puritan Bennett 840 ventilator (Medtronic; Carlsbad, California, USA), with the following settings: tidal volume of 6 mL/Kg predicted body wight, plateau pressure ≤27 cmH2O, and driving pressure ≤15 cmH2O. After 30 min of ventilation in a supine positioning the ventilatory variables, including perfusion indexes and anaerobic metabolism, were assessed. The arterial and central venous blood gases were determined in the GEM® PREMIER™4000 with iQM® equipment. 

The propofol infusion was administered to maintain a Richmond Agitation-Sedation Scale (RASS) score of -3 (moderate sedation; patient had any movement in response to voice, but no eye contact) and overcome ventilator asynchrony, obtain a level of awake sedation, optimizing the patient´s respiratory status without effects on respiratory pattern, respiratory drive, and arterial and central venous blood gases.”

Discussion

6. An interesting part that the authors could also analyze is the presence of significance between live and dead with respect to creatinine and bases. Which is well seen in the comparison between the two groups, then in the Cox Regression the creatinine does not become significant but the bases remain significant. Did the authors think that the renal part might also be involved in these mechanisms?

Reply: Although creatinine and bases levels showed significant differences between the non-survivors and survivors (p< 0.05), only the base was statistically significant in the univariate Cox regression analysis. However, according to the multivariate Cox regression analysis the base had no significant association with the non-survivors. For this reason, we do not consider that the renal part might also be involved in these mechanisms. This response was included in the new submission (Discussion Section, Page 9, paragraph 3).

7. One thing that has not been taken into consideration by the authors is also the fact that patients are affected by hypoxia, and thus come to the hospital, after a period when the hemoglobin dissociation curve (p50) has shifted, allowing patients to better "tolerate" hypoxia as shown by many present studies. This aspect could also influence the outcome of the work. What do the authors think about this aspect? Did they take it into consideration?

Reply: We agree with the reviewer. An important point is that the deterioration of lung function during SARS-CoV-2 infection induces alternative compensation mechanisms for oxygen uptake, such as the enhanced hemoglobin oxygen through a left shift of the oxygen dissociation curve, increasing perfusion, which modulates central venous blood gases [1]. Although the oxygen dissociation curve was not performed in this study, clinical relevance cannot be excluded. It needs further evaluation to determine their impact on the ∆Pv-aCO2/∆Ca-vO2 ratio in the prognostic of COVID-19 patients.

This response was included in this new submission in the Discussion Section (Page 11, paragraph 2).

Conclusion

8. I would be slightly softer, not stating that the ratio is an independent risk factor, but stating that in this study the ratio was found to be an independent risk factor, precisely also in light of the limitations of the study.

Reply: We made the correction in the conclusion as suggested by the reviewer:

In this study the ∆Pv-aCO2/∆Ca-vO2 ratio >2.14 mmHg/mL was an independent risk factor for mortality (HR 1.17, 95% CI 1.06-1.29, p= 0.001) in patients with severe ARDS secondary to SARS-CoV-2.

Thank you very much for your comments. Please let me know about above questions.

By signing this letter, we acknowledge that all the authors participated sufficiently to take public responsibility for its content. All of the authors have given their consent for submission to the journal. Further, we have no commercial associations which impact this work.

Best regards, 

Professor José Manuel Reyes 

Mexican Social Security Institute (IMSS)

E-mail: jose.reyesr@imss.gob.mx

Reference

1. Böning D, Kuebler WM, Bloch W. The oxygen dissociation curve of blood in COVID-19. American Journal of Physiology-Lung Cellular and Molecular Physiology. 2021;321: L349–L357. doi:10.1152/ajplung.00079.2021

---

## [Decision Letter · Decision Letter 3]

7 Aug 2023

The ∆Pv-aCO2/∆Ca-vO2 ratio as a predictor of mortality in patients with severe acute respiratory distress syndrome related to COVID-19

PONE-D-22-29034R3

Dear Dr. Reyes-Ruiz,

We’re pleased to inform you that your manuscript has been judged scientifically suitable for publication and will be formally accepted for publication once it meets all outstanding technical requirements.

Kind regards,

Samuele Ceruti

Academic Editor

PLOS ONE

Additional Editor Comments (optional):

Reviewers' comments:

Reviewer's Responses to Questions

**Comments to the Author**

1. If the authors have adequately addressed your comments raised in a previous round of review and you feel that this manuscript is now acceptable for publication, you may indicate that here to bypass the “Comments to the Author” section, enter your conflict of interest statement in the “Confidential to Editor” section, and submit your "Accept" recommendation.

Reviewer #3: All comments have been addressed

Reviewer #4: All comments have been addressed

2. Is the manuscript technically sound, and do the data support the conclusions?

Reviewer #3: Yes

Reviewer #4: (No Response)

3. Has the statistical analysis been performed appropriately and rigorously? 

Reviewer #3: Yes

Reviewer #4: (No Response)

4. Have the authors made all data underlying the findings in their manuscript fully available?

Reviewer #3: Yes

Reviewer #4: (No Response)

5. Is the manuscript presented in an intelligible fashion and written in standard English?

Reviewer #3: Yes

Reviewer #4: (No Response)

6. Review Comments to the Author

Reviewer #3: I thank the authors for the revision and the timely answers given to my questions and doubts.

The work now seems to me more precise and timely, ensuring a precise and timely logical and scientific thread.

A few small details still to be redefined, such as the missing reference in the RASS, you may choose ( Sessler CN, Grap MJ, Brophy GM. Multidisciplinary management of sedation and analgesia in critical care. Semin Respir Crit Care Med. 2001;22(2):211-26. doi: 10.1055/s-2001-13834. PMID: 16088675. ; Sessler CN, Gosnell MS, Grap MJ, Brophy GM, O'Neal PV, Keane KA, Tesoro EP, Elswick RK. The Richmond Agitation-Sedation Scale: validity and reliability in adult intensive care unit patients. Am J Respir Crit Care Med. 2002 Nov 15;166(10):1338-44. doi: 10.1164/rccm.2107138. PMID: 12421743. ; Ely EW, Truman B, Shintani A, Thomason JW, Wheeler AP, Gordon S, Francis J, Speroff T, Gautam S, Margolin R, Sessler CN, Dittus RS, Bernard GR. Monitoring sedation status over time in ICU patients: reliability and validity of the Richmond Agitation-Sedation Scale (RASS). JAMA. 2003 Jun 11;289(22):2983-91. doi: 10.1001/jama.289.22.2983. PMID: 12799407. )

And a key point that doesn't add up in the methods. You state to me that the study was conducted 30 minutes after intubation of the patient, in the supine position, thus in complete rest of the patient. Consequently, propofol infusion was not indicated for a RASS of -3 at that time. It would be to specify that SUCCESSFULLY I imagine you set a goal of RASS -3 for the reasons described in the sentence. This point should be specified further so as not to fall into confusion in reading the paper. I think it is merely an issue of language and adequate description of the condition.

Thank you again for the accurate answers to my questions.

Reviewer #4: (No Response)

7. PLOS authors have the option to publish the peer review history of their article (what does this mean?). If published, this will include your full peer review and any attached files.

Reviewer #3: No

Reviewer #4: **Yes: **Mumoli Nicola, MD - Department of Internal Medicine, Magenta Hospital, Magenta (MI), Italy

---

## [Editor Report · Acceptance letter]

11 Aug 2023

PONE-D-22-29034R3 

The ∆Pv-aCO_2_/∆Ca-vO_2_ ratio as a predictor of mortality in patients with severe acute respiratory distress syndrome related to COVID-19 

Dear Dr. Reyes-Ruiz:

I'm pleased to inform you that your manuscript has been deemed suitable for publication in PLOS ONE. Congratulations! Your manuscript is now with our production department. 

Kind regards, 

on behalf of

Dr. Samuele Ceruti 

Academic Editor

PLOS ONE